# An Improved Approach to Wave Energy Resource Characterization for Sea States with Multiple Wave Systems

Xingjie Jiang [1,2,3], Dalu Gao [1,2,3], Feng Hua [4], Yongzeng Yang [1,2,3,*] and Zeyu Wang [1,2,3]

1 First Institute of Oceanography, and Key Laboratory of Marine Science and Numerical Modelling, Ministry of Natural Resources, Qingdao 266061, China
2 Laboratory for Reginal Oceanography and Numerical Modelling, Pilot National Laboratory for Marine Science and Technology, Qingdao 266237, China
3 Shandong Key Laboratory of Marine Science and Numerical Modelling, Qingdao 266061, China
4 College of Science, Shantou University, Shantou 515063, China
* Correspondence: yangyz@fio.org.cn

**Abstract:** Generally, wave energy resource assessment and characterization are performed based on an entire wave spectrum, ignoring the detailed energy features that belong to wave systems, i.e., wind waves and swells. In reality, the energy is separately possessed by multiple wave systems, propagating at different directions and velocities. Therefore, it is the wave system that is the most fundamental unit of the wave energy resource. Although detailed analyses of wind waves and swells can be conducted via wave system partitioning, operational assessment methods that can reveal the detailed wave energy characteristics of wave systems still deserve further development. Following a two-step partitioning procedure, this paper presents an improved approach to the characterization of wave energy resources based on grouped wave systems. Wave systems classified as the same group are consistent in terms of propagation direction, velocity, and other characteristics of wave energy, but these characteristics between the groups are obviously different. Therefore, in comparison with the traditional method, the new approach can reveal more comprehensive and more detailed characteristics of the wave energy resource in terms of (i) wind-sea and swell components, (ii) directionality, and (iii) wave conditions; details that represent valuable information for the improvement of the performance of wave energy converter devices and the optimization of the layout of device arrays in wave farms.

**Keywords:** wave energy characterization; spectral partition; wave environment; wave modeling

## 1. Introduction

In response to projections regarding future climate change and the gradual depletion of fossil energy sources, global attention is becoming focused on harnessing renewable energy. The various sources of marine renewable energy, which include offshore wind energy, wave energy, and tidal energy, are of particular interest because of their abundant reserves and environmentally friendly properties. Wave energy is one of the most concentrated [1] and highly available sources of marine renewable energy with great potential for exploitation. Primary considerations in harnessing such a resource are its assessment and characterization, which relate to the siting of wave farms, the design of wave energy converter (WEC) devices, and the layout of WECs in farms.

Generally, the assessment and characterization of wave energy resources are highly correlated with the wave climatology of the target area, focusing on annual or seasonal spatial distributions and the temporal variability of local wave energy characteristics, as well as the reserve of such resources. However, recent research suggests that wave energy flux with $J > 2\,\mathrm{kW/m}$ can be considered exploitable [2–4], and such wave conditions can be achieved in most cases. Moreover, there is a rated power, that is, an upper limit on the output power of all WECs (the rated power of a selection of WECs can be found

in [5]). Therefore, the actual amount of exploitable energy ultimately depends more on the WEC performance than on the entire wave energy potential [5–9]. Therefore, more detailed characterization must be performed at sites of interest regarding features such as:

(i) The wind-sea and swell components comprised in the resource potential. The distinct characteristics presented by wind-driven seas and swell, especially the different spectral (band) width and directional spreading, can substantially affect WEC performance. Specifically, WECs are sensitive to spectral width, which is highly related to the groupness of the sea state, and this sensitively is found to be more pronounced when the response band of the device is broad [10]. Moreover, certain WECs are also sensitive to the direction of energy propagation, whereby broad directional spreading might reduce the expected energy output.

(ii) The propagation direction of the wave energy. As mentioned above, the deployment of certain WECs, e.g., the Wavestar [11], Pelamis [12], and Wave Dragon [13], requires that the WECs be fixed with respect to the prevailing wave direction in the locality of the farm, and they cannot take full advantage of wave trains coming from other directions. Even though some WECs, e.g., Ceto [14] and AquaBuOY [15], involving the active part moving on a vertical axis, are insensitive to the propagation direction of waves, the optimization of such WEC arrays must also consider the propagation direction of waves to avoid WEC members in those arrays being affected by the 'shadows' of the others and fully use the wave energy resource in the prevailing direction.

(iii) The wave conditions. The performance of WECs can also depend markedly on wave conditions, including both drifting and no-drifting floating devices [16]. The wave conditions generally exhibit as bivariate distributions of significant wave height $H_s$ and wave energy period $T_e$ (expressed as the ratio of $(-1)$-th to the 0-th spectral moment). The conversion efficiency and electricity production of WECs under different combinations of $H_s$ and $T_e$, denoted as the "power matrix" [5–7,9], can be provided by the WEC manufacturers (examples of the power matrix of a selection of WECs can be found in the appendix of [6]). Moreover, for wave energy characterization, the $H_s - T_e$ scatter table can be considered a "fingerprint" that provides detailed information regarding the long-term wave environment at a specific location.

In the assessment and characterization of the wave energy resource, most of the wave energy characteristics (some of them are listed in Section 2.3), including those that represent the three aspects discussed above, can be calculated through analysis of two-dimensional (2D) wave spectra. The parameters are generally obtained by integrating throughout an entire spectrum, without considering the detailed energy features that belong to wave systems, i.e., wind-sea and swells (examples can be found in the recent publications [17–24]). However, in reality, the wave field at any given time and place is likely to be composed of a collection of wave systems. The wave energy contained in such a wave field is separately possessed by these systems, propagating with different velocities in different directions. Therefore, it is the wave system that is the most fundamental unit of the wave energy resource. Although the Technical Specification proposed by the International Electrotechnical Commission (IEC TS 62600-101:2015; hereafter, IEC2015) [25] has recommended employing wave spectral partitioning in the energy resource analyses, and also some previous research, e.g., [26–29], have discussed wave energy characteristics in the forms of wind waves and swells separately, they failed to form an operational approach to characterizing the energy features of wave systems under the current framework of the wave energy-harnessing industry.

This paper proposes an improved approach to the assessment of wave energy resources in multiple wave system-coexisting wave fields. In the approach, the wave systems (i.e., the large numbers of wave spectral partitions mentioned above) are grouped, and the widely accepted characteristic parameters of wave energy can then be calculated and counted based on the grouped wave systems. To demonstrate the newly proposed approach, an experimental case is considered, in which a long time series of simulated wave spectra are partitioned and grouped before the characteristic parameters are calculated and counted.

For comparison, statistics of those same parameters obtained from the same series of wave spectra but without partitioning are also presented.

The remainder of this paper is organized as follows. The methods involved in the new approach, including the acquisition of the long time series of wave spectra, partitioning and grouping procedure, and the determination of the characteristic parameters of wave energy, are all introduced in Section 2. In Section 3, the statistics of the key parameters obtained from both the partition-grouped and the nonpartitioned wave spectra are illustrated, and relevant comparisons and elucidations are discussed. Finally, a discussion and the derived conclusions are presented in Sections 4 and 5, respectively.

## 2. Materials and Methods

### 2.1. Modeling of Long Time Series Wave Spectra

To demonstrate the new approach to the characterization of wave energy, hourly wave field modeling of the Angola offshore area (West Africa) was conducted for a 20-year period (2000–2019). According to the West Africa Swell Project [30–32], swell generated by storms in the region 40°–60° S in the South Atlantic can approach the Angola offshore area from the S to SW direction, with a sea state of up to 4 m, and swell generated by large storms moving off North America may travel across the North and South Atlantic, approaching the target area from the NW direction with relatively small energy (<0.5 m). Under the influence of the southeast trade winds of the South Atlantic, S–SE winds prevail over the Angola offshore area throughout the year. Moreover, equatorial westerlies [33,34] can also occur occasionally in the equatorial trough over the Atlantic. The significant swell and the effects of local trade winds and equatorial westerlies mean that the Angola offshore area is highly suited to the exhibition of various wave systems.

Third-generation wave models, e.g., SWAN [35,36], WaveWatch III [37–39], etc., are widely used in wave energy resource assessment and characterization research. These models can directly simulate the 2D wave spectra and generate long-term wave parameters with continuous coverage in space and time, which can be applied in further analyses. In this study, the MASNUM-WAM [40–42] (formerly LAGFD-WAM) was adopted to perform the 20-year modeling of the wave fields. MASNUM-WAM is a third-generation wave model which solves the energy spectrum balance equation using a characteristic inlaid scheme [41]; the white-capping dissipation of this model is based on the wave broken statistic theory [43], the wind input source term is adopted from [44], and the DIA [45,46] scheme is adopted to calculate the nonlinear energy transfer between waves. MASNUM-WAM has been widely adopted in scientific and engineering research (e.g., [47–50]). Moreover, MASNUM-WAM is now the ocean wave component of several operational ocean forecasting systems (OFS), such as the OFS for the seas of China and adjacent areas [51], the OFS for Southeast Asian Seas, and the OFS for the 21st-Century Maritime Silk Road [52].

As illustrated in Figure 1a, a pair of nested computational grids was used in the simulation. The parent grid, named WA-outside, covered the region $-80°$ N to $10°$ N, $-110°$ E to $15°$ E with $0.5° \times 0.5°$ horizontal resolution. The child grid, named WA-inside, covered the region $-37°$ N to $9°$ N, $-20°$ E to $14°$ E with $0.125° \times 0.125°$ horizontal resolution. With consideration of the swell from the South Atlantic, the southern and western borders of WA-outside extended to the Antarctic continent, and $-110°$ E, respectively. However, given that the NW swell is ignorable, the northern border of WA-outside was set to $10°$ N, and as a boundary condition, artificial spectra, that is:

$$E(f, \theta) = F(f)D(\theta) \tag{1}$$

was continuously input from the northern border during the modeling. In Equation (1), $F(f)$ is the JONSWAP spectrum [53]:

$$F(f) = \alpha \frac{g^2}{(2\pi)^4} f^{-5} exp\left[ -\frac{5}{4} \left( \frac{f}{f_p} \right)^{-4} \right] \gamma_J^{exp[-\frac{1}{2F(f)^2} \left( \frac{f}{f_p} -1 \right)^2]} \tag{2}$$

with peak enhancement $\gamma_J = 3.3$ and parameter $\beta = \begin{cases} 0.07, f \leq f_p \\ 0.09, f > f_p \end{cases}$; both the Philips constant $\alpha$ and the peak frequency $f_p$ were obtained according to the wind speed at the border with a 15 km fetch. The directional distribution function in Equation (1) is designed as follows:

$$D(\delta) = \frac{2}{\pi}cos^2(\delta) \tag{3}$$

where $\delta$ denotes the angle between the direction in the spectrum and the direction in which the wind is blowing. In this study, the modeling spectral space was set as 24 directions with intervals of 15° and 35 frequencies spaced logarithmically from the minimum frequency of 0.042 Hz up to 1.07 Hz with intervals of $f_{i+1}/f = 1.1$. Moreover, the 2D wave spectra simulated at six sites, named P01–P06 (see Figure 1b and Table 1), were selected to demonstrate the new characterization approach.

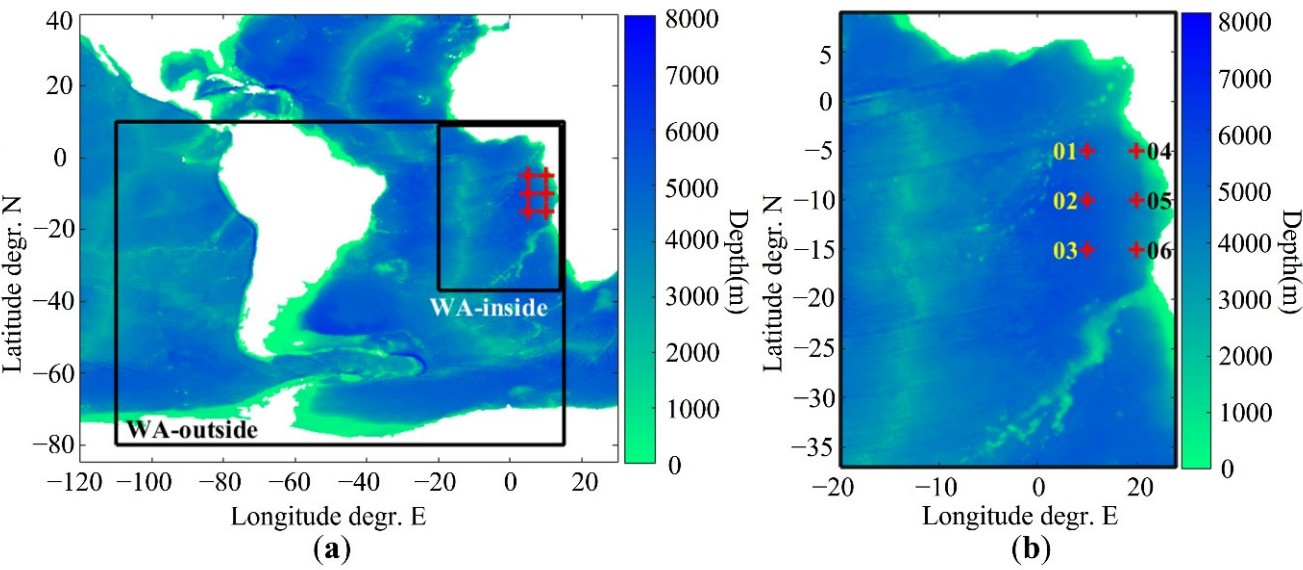

**Figure 1.** Nested computational grids and locations of sites P01–P06: (**a**) the parent and child grids; (**b**) locations of P01–P06 in the child grid.

**Table 1.** Locations of sites P01–P06.

| SiteID | Latitude (°N) | Longitude (°N) | Depth (m) |
|--------|---------------|----------------|-----------|
| P01 | −5 | 5 | 4949 |
| P02 | −10 | 5 | 5423 |
| P03 | −15 | 5 | 5488 |
| P04 | −5 | 10 | 2929 |
| P05 | −10 | 10 | 4294 |
| P06 | −15 | 10 | 3798 |

The wind force adopted in the simulations was derived from ECMWF-ERA5 reanalysis hourly data (DOI: 10.24381/cds.adbb2d47), which provided the u–v wind field at the height of 10 m above the sea surface with 0.25° × 0.25° horizontal resolution. Bathymetric data were obtained from ETOPO1 of NGDC (DOI:10.7289/V5c8276m), and the shoreline data were obtained from the GSHHG [54].

Although the wave field modeling might be considered not very rigorous, the purpose of this paper is notably to elucidate the new approach to wave energy characterization rather than to assess the wave energy resource in the Angola offshore area. A simple validation of the simulated results was performed by comparing the simulated $H_s$ with the observations of the Jason-2 satellite in 2016. The simulated results are gridded in the spatial-temporal domain, while the satellite observations are a series of times and locations along

the satellite tracks. To make them comparable, the simulated $H_s$ were interpolated to the time and location where each observed record was obtained. The $H_s$ differences (observed minus simulated-interpolated) along the Jason-2 tracks are shown in Figure 2a, where most of the differences are within $[-0.5\ 0.5]$ meters as denoted by the colors; another comparison of the simulated and observed $H_s$ is expressed based on 0.05 m × 0.05 m wave height cells as shown in Figure 2b, where the color in each cell indicates the number of occurrences of the simulated-observed $H_s$ pairs. The mean absolute error and the correlation coefficient between the observed and simulated-interpolated $H_s$ are 0.27 m and 0.84, respectively. The illustrations above reveal an acceptable reproduction of the observed sea states; therefore, the simulation can provide a reliable foundation for the following demonstrations.

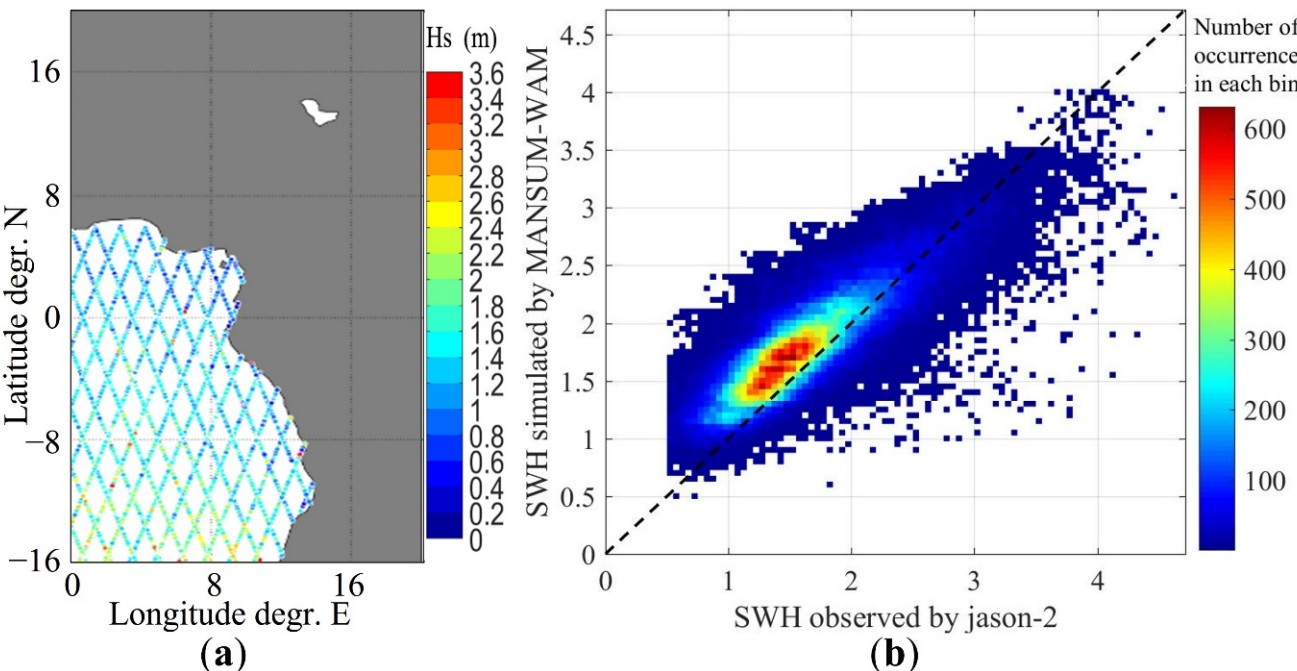

**Figure 2.** Validation of the simulated $H_s$: (**a**) $H_s$ differences (observed minus simulated) along the satellite tracks in the Angola offshore area; the colors indicate the values of differences in meters; (**b**) scatter diagram of the simulated and observed $H_s$; the colors indicate the number of occurrences of the simulated-observed $H_s$ pairs in the 0.05 m × 0.05 m wave height cells.

*2.2. Partitioning and Grouping Procedure*

The spectral partitioning technique can be traced back to a digital image processing watershed algorithm [55], which can be adopted to identify watershed lines, mountain peaks, and valleys in topographic maps. Because the 2D spectrum (see Figure 3) resembles a topological surface, it is logical to apply such an algorithm in this circumstance [56]. Figure 3a, which is taken from Figure 1 of [57], shows the basic approach to the spectral partitioning method, that is, by searching through the spectral matrix $S(f, \theta)$, the paths of steepest ascent leading to each peak, or the local energy maximum (denoted by the large numbers in Figure 3a) can be identified; then, all paths leading to the same peak can be grouped, and the members that lie on the collection of the paths are considered to belong to a distinct partition. In Figure 3b, the $f - \theta$ domain has been divided into three partitions labeled 1–3, and the boundaries of the partitions are identified by white lines. Each partition contains a 'peak' that has the highest spectral density among the surroundings, and that (together with its surroundings) is considered representative of the energy from one of the subsystems within the spectrum. The partitioning of wave spectra is widely adopted in research concerning data assimilation (e.g., [56,58,59]), spatial and temporal tracking of wave systems (e.g., [57,60]), and so on.

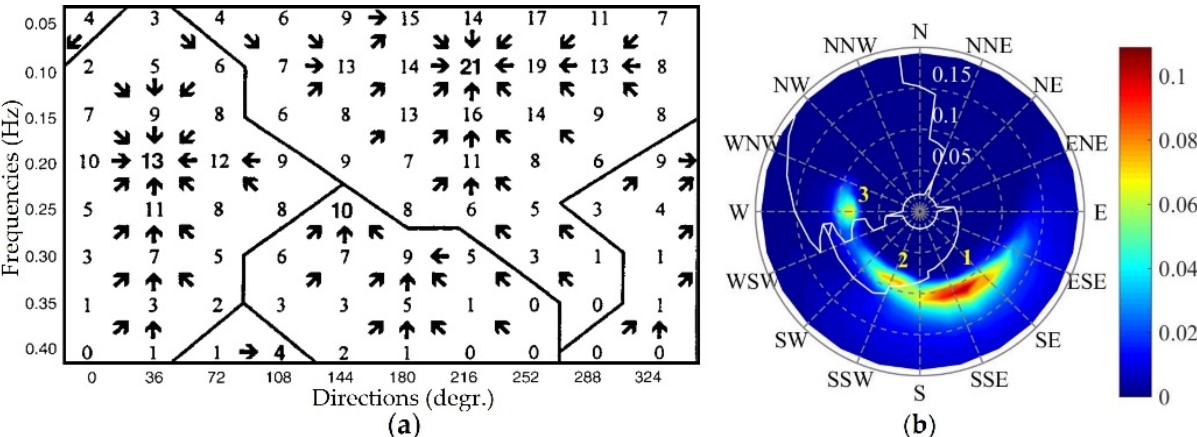

**Figure 3.** Demonstration of spectral partitioning method: (**a**) illustration of the basic approach to spectral partitioning method (Reprinted with permission from Figure 1 of Ref. [57] Copy right 2001, the American Meteorological Society); the numbers indicate spectral density; (**b**) An example of the partitioning of two-dimensional wave spectrum: the $f - \theta$ spectral domain is demonstrated in a polar coordinate, the spectral density is indicated by the colors, the partitions are labeled 1–3, and the boundaries of the partitions are identified by white lines.

Portilla-Yandún et al. [61–63] proposed a two-step partitioning procedure: by partitioning each 2D wave spectrum in a long time series (first-step partitioning), individual wave systems contained in each spectrum can be identified; the peaks of the identified wave systems could occur at any position in the $f - \theta$ spectral space, and the probability of their occurrence then forms another 2D spectrum; by partitioning the newly formed spectrum (second-step partitioning), the identified wave systems can then be grouped automatically. It is apparent that systems classified as the same group have similar peak frequencies and directions; thus, the energy contained in each group naturally shares similar propagation velocity and direction. Furthermore, for wave systems at the same spatial point, their similar status of propagation indicates similar generation patterns: for swell systems, they may be formed and propagate from the same point of origin; for wind wave systems, they may be generated by the same relatively stable local meteorological pattern, such as monsoons or trade winds. Therefore, the statistics of grouped systems over a long time series clearly reflect physical reality and have climatic significance.

In this study, the concept of "two-step partitioning" is applied to wave energy characterization. The partitioning program implemented was developed based on the W3PARTMD module of WaveWatchIII ver. 6.07 [64]), which can be traced back to the MATLAB code [65,66] that was used to apply the watershed algorithm [55]. In the newly developed partitioning program, the partitions are identified by the basic approach shown in Figure 3a, without further identification of wind-sea/swell systems or any combination of those systems. Furthermore, parameters representing the characteristics of wave energy (see Section 2.3) can be calculated directly from the identified partitions in the program.

For the six selected sites in the Angola offshore area, firstly, each of the spectra simulated during the modeling was segmented into one or several partitions, i.e., wave systems, and the corresponding characteristic parameters (see Section 2.3) were calculated simultaneously; secondly, at each site, the peaks of the identified systems were automatically assigned to the 35 × 24 frequency-direction cells in the modeling spectral space, and the number of the peak occurrences formed a new 2D spectrum in the frequency-direction domain; finally, the partitioning program was applied again to the newly formed 2D spectrum again, and those identified wave systems at each selected site were grouped. Further wave energy characterization then can be performed based on those groups, in conjunction with the corresponding characteristic parameters mentioned above.

### 2.3. Characteristic Parameters of Wave Energy

The parameters mentioned in this study are all referred to IEC2015 [25], except for the parameter of the "wind-sea fraction." It is noted that these parameters can be calculated by integrating both the identified partitions and the nonpartitioned spectra.

1.  Wave spectra and spectral moments

The 2D wave spectrum (or partition thereof) can be expressed as $S(f, \theta)$, in which $f$ denotes the frequency in Hertz (Hz) and $\theta$ denotes the direction in radians or degrees (°). The one-dimensional spectrum $S(f)$ can be obtained as follows:

$$S(f) = \int_0^{2\pi} S(f, \theta) \, d\theta \tag{4}$$

and the $n$-th spectral moment can be expressed as below:

$$m_n = \int_0^{\infty} f^n S(f) \, df \tag{5}$$

In the $f - \theta$ spectral space, the location $(f_p, \theta_p)$ with the maximum spectral density is identified as the spectral peak.

2.  Significant wave height and wave energy period

The significant wave height is calculated as follows:

$$H_s = 4\sqrt{m_0} \tag{6}$$

and the wave energy period is expressed as below:

$$T_e = \frac{m_{-1}}{m_0} \tag{7}$$

In Equations (6) and (7), $m_0$ and $m_{-1}$ denote the $0$-th and $-1$-th spectral moments, respectively.

3.  Omnidirectional wave energy flux

The omnidirectional or directionally unresolved wave energy flux characterizes the time-averaged energy flux through an envisioned vertical cylinder of unit diameter; its unit is usually taken as kilowatts per meter (kW/m). This parameter can be estimated as follows:

$$J = \rho g \int_0^{2\pi} \int_0^{\infty} C_g(f, d) S(f, \theta) \, df \, d\theta \tag{8}$$

where $\rho$ (which is taken as 1023 kg/m$^3$ in this study) denotes the density of seawater, $g = 9.81$ m/s$^2$ is the acceleration of gravity, and $C_g$ is the group velocity of waves, which is expressed as below:

$$C_g = 2\pi \frac{\partial f}{\partial k} = \frac{2\pi f}{k} \left( 1 + \frac{2kd}{\sinh(2kd)} \right) \tag{9}$$

In the equation above, wave number $k$ associated with given frequency $f$ and water depth $d$ (unit: m) is defined implicitly through the dispersion relation:

$$(2\pi f)^2 = gk\tanh(kd) \tag{10}$$

Parameter $J$ can also be calculated approximately using $H_s$ and $T_e$:

$$J \approx \frac{\rho g}{16} H_s^2 C_g(T_e, d) \tag{11}$$

In deep water, where $kd \gg 1$, parameter $J$ can be expressed more simply as follows:

$$J = \frac{\rho g^2}{64\pi} H_s^2 T_e \tag{12}$$

In this paper, parameter $J$ is obtained using Equation (8) except where specified.

4.  Directionally resolved wave energy flux

The omnidirectional wave energy flux can be resolved to direction $\theta_j$, and this parameter can be obtained as follows:

$$J_\theta = \rho g \int_0^{2\pi} \int_0^\infty C_g(f,d) S(f,\theta) \cos(\theta - \theta_j) \delta d f d\theta, \quad \begin{cases} \delta = 1, \cos(\theta - \theta_j) \geq 0 \\ \delta = 0, \cos(\theta - \theta_j) < 0 \end{cases}. \tag{13}$$

The resolved $J_\theta$ represents the time-averaged energy flux through an envisioned vertical plane of unit width that has its normal vector parallel with $\theta_j$. The direction corresponding to the maximum value of $J_\theta$, denoted here as $J_{\theta_{Jmax}}$, is defined as "the direction of maximum directionally resolved wave energy flux" ($\theta_{Jmax}$). To measure the directional spread of wave power, the directionality coefficient $d_\theta$ is adopted, which is calculated as the ratio of the maximum directionally resolved $J_{\theta_{Jmax}}$ to the omnidirectional $J$:

$$d_\theta = \frac{J_{\theta_{Jmax}}}{J} \tag{14}$$

In this study, because the spectral space of the simulated spectra is divided into 24 directions (see Section 2.1), parameters $\theta_{Jmax}$ and $d_\theta$ are selected from the 24 $J_\theta$ values for each spectrum/partition.

5.  Spectral width

Spectral width characterizes the relative spread of energy in the $f$ direction of the wave spectrum/partition. Parameter $\epsilon_0$, which represents the spectral width, can be expressed as follows:

$$\epsilon_0 = \sqrt{\frac{m_0 m_{-2}}{m_{-1}^2} - 1} \tag{15}$$

where $m_0$, $m_{-2}$, and $m_{-1}$ denote the $0$-th, $-2$-th, and $-1$-th spectral moments, respectively.

6.  Wind-sea fraction

The wind-sea fraction parameter $W$ is introduced to characterize the proportion of wind-wave energy contained in each spectrum/partition. Following Hanson and Phillips [57] and Tracy et al. [67], parameter $W$ is calculated as follows:

$$W = E^{-1} E_{U_p > c} \tag{16}$$

where $E = m_0 = \int_0^{2\pi} \int_0^\infty S(f,\theta) d f d\theta$ is the total spectral energy, and $E_{U_p > c}$ is the energy in the spectral space for which the projected wind speed $U_p$ is larger than the local wave phase velocity $c$. The projected wind speed can be calculated as follows:

$$U_p = C_{mult} U_{10} \cos(\delta) \tag{17}$$

where $U_{10}$ denotes wind speed at the height of 10 m above the sea surface, $\delta$ denotes the angle between the direction in the spectral space and the direction in which the wind is blowing, and $C_{mult}$ is a coefficient. In the spectral space, wave phase velocity $c$ is associated with frequency $f$ and wave number $k$; thus, according to Equation (10):

$$c = \frac{2\pi f}{k} = \sqrt{\frac{g}{k}} \cdot \sqrt{\tanh(kd)} \tag{18}$$

In the spectral space, the condition $U_p > c$ actually defines an area that is under the direct influence of the wind, and the coefficient $C_{mult}$ is introduced to allow fully grown wind seas within this area. In this paper, coefficient $C_{mult}$ is taken as 1.7 in accordance with the default value set in the WaveWatchIII model [68].

## 3. Results

### 3.1. Wave System Groups

The grouped wave systems at the selected sites are shown in Figure 4. In each panel of Figure 4, the frequency region shown is restricted to 0–0.2 Hz, and the directions indicate the origin of wave energy propagation; the colors in the spectral space denote the number of occurrences of the identified peaks, and the groups are labeled 1–3 (hereafter, grp-1, grp-2, and grp-3), with their boundaries represented by white lines. The accumulated wave energy flux $J$ (Equation (8)) and the number of systems contained in each group at each site are also listed in Table 2, and both parameters are presented as yearly averages.

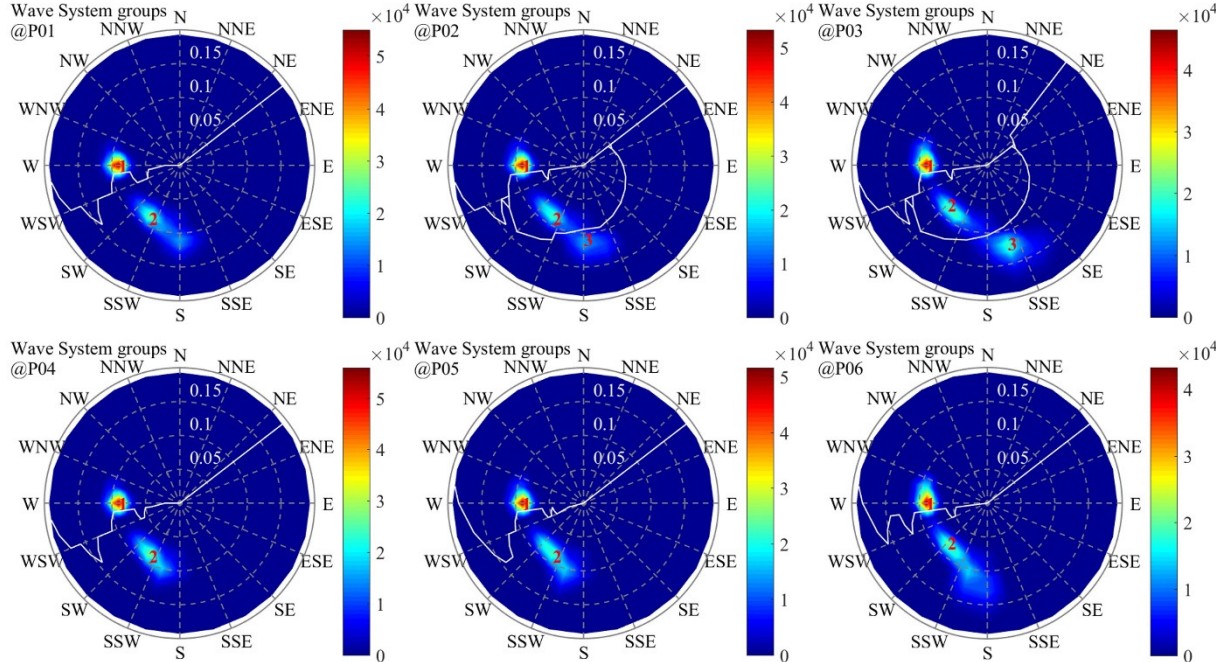

**Figure 4.** Peak occurrence spectra and grouped wave systems at sites P01–P06. The number of occurrences of the identified peaks is indicated by the colors. The classified groups are labeled 1–2 or 1–3, and the boundaries of the groups are identified by white lines.

**Table 2.** Statistics of wave energy flux ($J$) at sites P01–P06.

| SiteID | Accumulated $J$ Contained in Each Group (Yearly Averaged, MW/m) | | | Number of Systems Occurring in Each Group (Yearly Averaged) | | |
|---|---|---|---|---|---|---|
| | grp-1 | grp-2 | grp-3 | grp-1 | grp-2 | grp-3 |
| P01 | 25.32 | 96.95 | - | 9382.85 | 14,593.35 | - |
| P02 | 34.74 | 91.98 | 27.11 | 9188.55 | 10,459.90 | 4748.95 |
| P03 | 45.05 | 106.36 | 47.53 | 8743.70 | 10,557.35 | 6256.60 |
| P04 | 29.48 | 93.40 | - | 9437.35 | 13,300.50 | - |
| P05 | 38.36 | 113.40 | - | 9089.20 | 13,194.15 | - |
| P06 | 46.91 | 154.80 | - | 8543.30 | 13,927.95 | - |

Figure 4 shows that two or three groups coexist at each of the six sites. Systems coming from the W direction, denoted as grp-1, can be found in all panels, and the peaks in this group occur with the greatest concentration. The peaks classified as grp-2 in each panel

occur most frequently at directions between SSW and SW; however, at P01 and P06, the grp-2 peaks are also slightly concentrated in the SSW–S direction. Independent groups, denoted as grp-3, can be found only at P02 and P03, and they are concentrated in the S–SSE direction. Additionally, Table 2 shows that the wave energy accumulated in grp-2 is most prominent at all six sites, and that it is the systems in grp-2 that occur most frequently. However, at sites P02 and P03, the total accumulated energy and the total number of occurrences of the systems in grp-1 and grp-3 can be similar to the values of grp-2. Finally, among all the groups, the $f_p$ values in grp-1 and grp-2 are similar, whereas the $f_p$ values in grp-3 are obviously higher.

### 3.2. Wind Wave and Swell Contributions to Wave Energy

The box-whisker plots [69–72] shown in Figure 5 illustrate the statistics of the wind-sea fraction obtained based on grp-1–grp-2 (or grp-1–grp-3) and all the nonpartitioned spectra (all) at sites P01–P06. The bottoms and tops of the boxes indicate the 25th ($Q_{25\%}$) and 75th ($Q_{75\%}$) percentiles, respectively, while the red lines inside the boxes indicate the sample median ($Q_{50\%}$). The distance between $Q_{25\%}$ and $Q_{75\%}$, that is, the length of the box, is the interquartile range ($IQR = Q_{75\%} - Q_{25\%}$), and the whiskers (dashed lines) above $Q_{75\%}$ and below $Q_{25\%}$ extend to $Q_{max} = Q_{75\%} + 1.5 \times IQR$ and $Q_{min} = Q_{25\%} - 1.5 \times IQR$, respectively. It is considered that samples within the range between $Q_{max}$ and $Q_{min}$ can represent the primary characteristics of all samples, while the samples outside that range might cause a shift in the overall characteristics; thus, these outliers are marked by red "+" symbols. Similar plots representing the distribution statistics of the spectral width $\epsilon_0$ and the directional coefficient $d_\theta$ can also be seen in Figures 6 and 7, respectively.

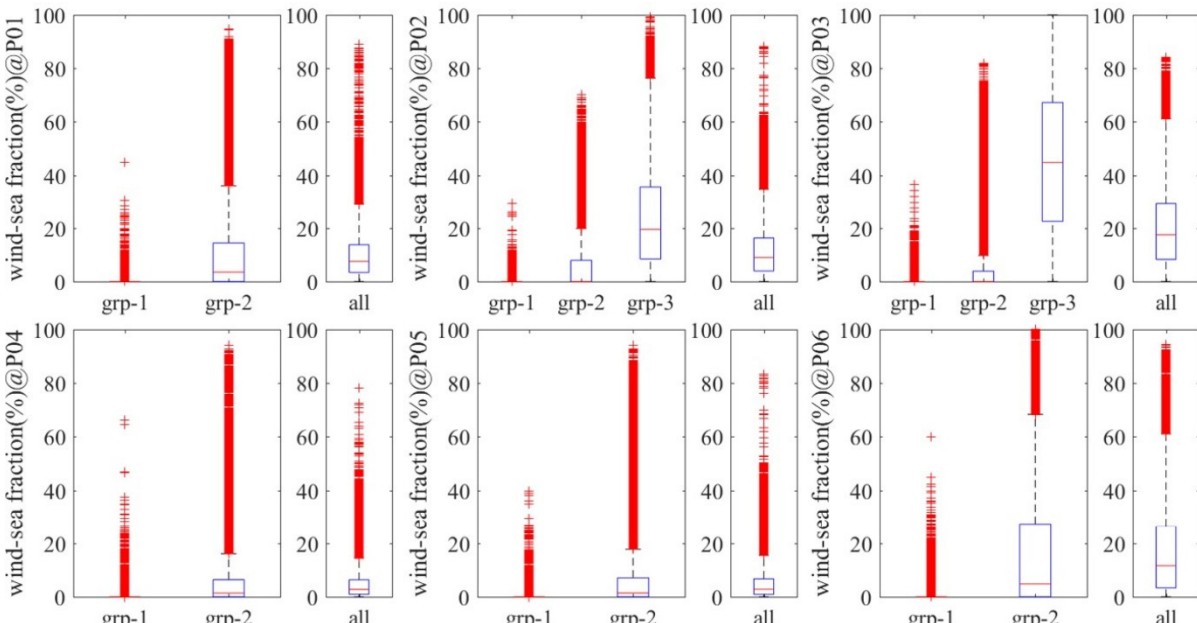

**Figure 5.** Box-whisker plots of wind-sea fraction at sites P01–P06. The bottoms and tops of the boxes indicate the 25th ($Q_{25\%}$) and 75th ($Q_{75\%}$) percentiles, respectively, while the red lines inside the boxes indicate the sample median ($Q_{50\%}$). The length of the box is the interquartile range ($IQR = Q_{75\%} - Q_{25\%}$), and the whiskers extend to $Q_{max} = Q_{75\%} + 1.5 \times IQR$ and $Q_{min} = Q_{25\%} - 1.5 \times IQR$, respectively. The samples outside the range between $Q_{max}$ and $Q_{min}$ are marked by red "+" symbols.

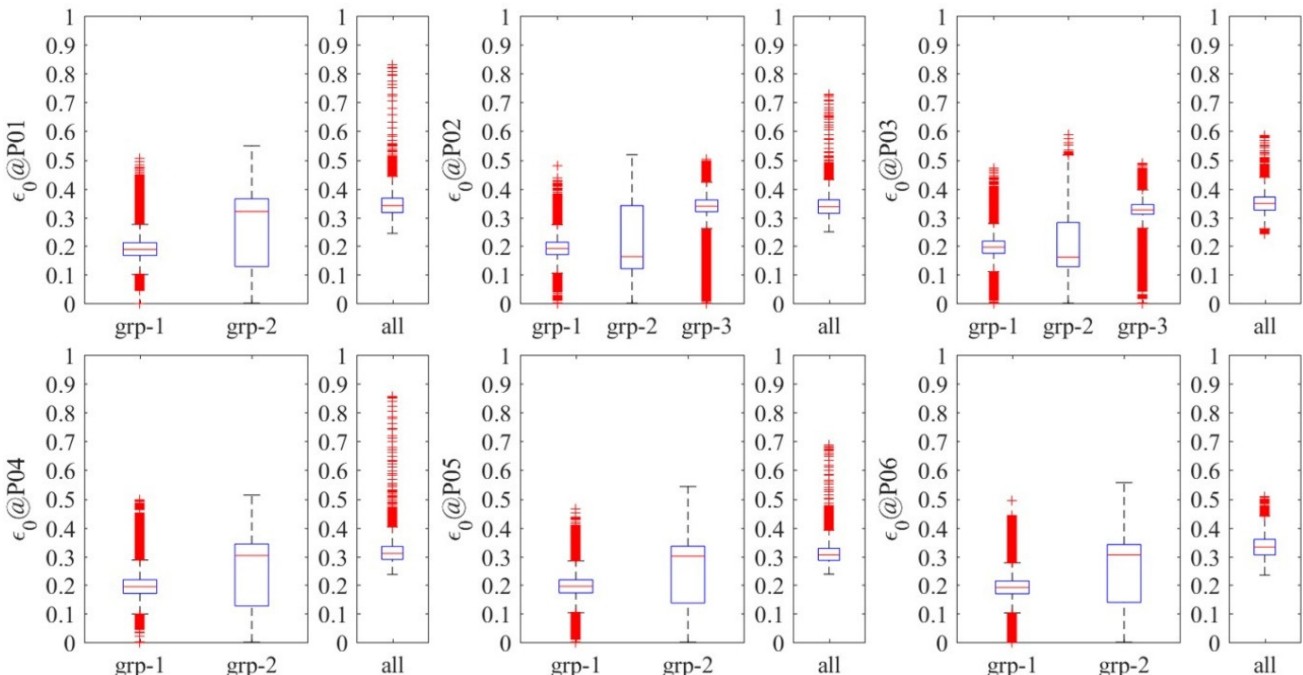

**Figure 6.** Box-whisker plots of spectral width $\epsilon_0$ at sites P01–P06. The bottoms and tops of the boxes indicate the 25th ($Q_{25\%}$) and 75th ($Q_{75\%}$) percentiles, respectively, while the red lines inside the boxes indicate the sample median ($Q_{50\%}$). The length of the box is the interquartile range ($IQR = Q_{75\%} - Q_{25\%}$), and the whiskers extend to $Q_{max} = Q_{75\%} + 1.5 \times IQR$ and $Q_{min} = Q_{25\%} - 1.5 \times IQR$, respectively. The samples outside the range between $Q_{max}$ and $Q_{min}$ are marked by red "+" symbols.

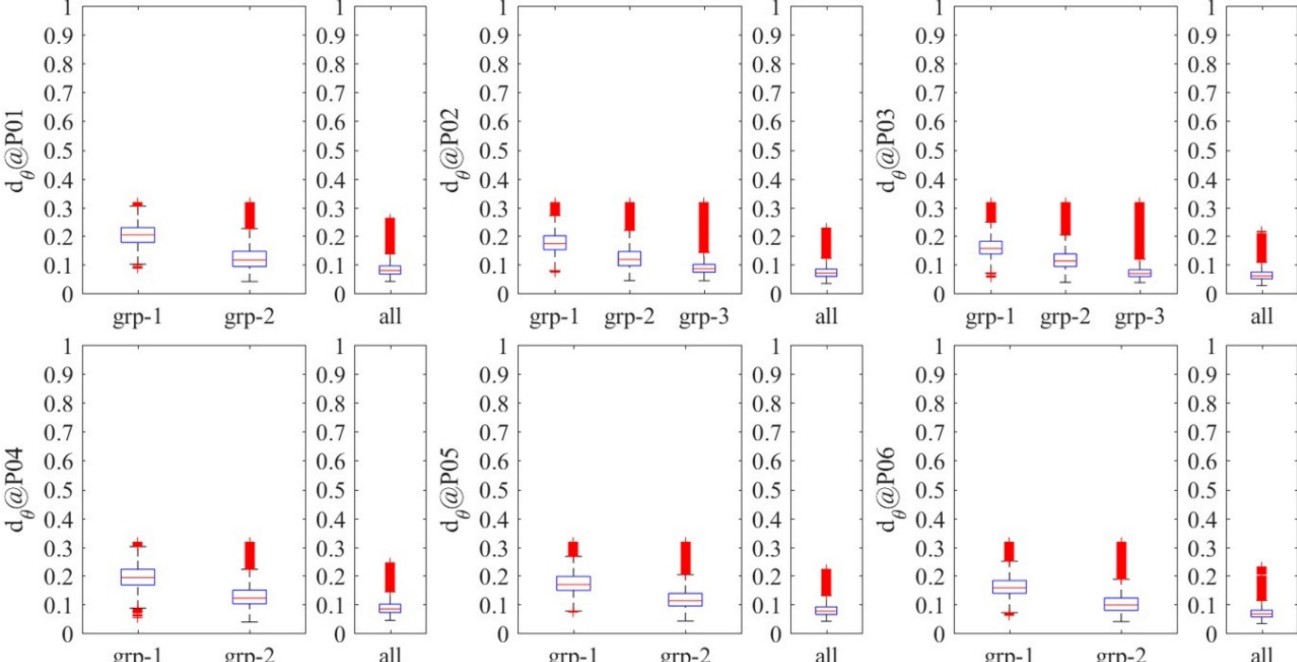

**Figure 7.** Box-whisker plots of directional coefficient $d_\theta$ at sites P01–P06. The bottoms and tops of the boxes indicate the 25th ($Q_{25\%}$) and 75th ($Q_{75\%}$) percentiles, respectively, while the red lines inside the boxes indicate the sample median ($Q_{50\%}$). The length of the box is the interquartile range ($IQR = Q_{75\%} - Q_{25\%}$), and the whiskers extend to $Q_{max} = Q_{75\%} + 1.5 \times IQR$ and $Q_{min} = Q_{25\%} - 1.5 \times IQR$, respectively. The samples outside the range between $Q_{max}$ and $Q_{min}$ are marked by red "+" symbols.

As defined by Equation (16) in Section 2.3, parameter $W$ tends toward 1 (0) for wind-seas (swell). It can be seen in Figure 5 that the systems contained in grp-1 at each site are almost all swell because the $Q_{max}$ values of $W$ in grp-1 are equal to zero. For grp-2, the distributions of $W$ are slightly different. At sites P02 and P03, although the outliers exceeding $Q_{max}$ are much higher than in grp-1, swell is still dominant because the $Q_{max}$ values are <10% and the $Q_{50\%}$ values are equal to 0. At sites P04 and P05, the outliers are even higher, and the sample medians are slightly larger than 0; nevertheless, it can be considered that the majority of energy contained in the two grp-2s is associated with swell. Conversely, at sites P01 and P06, the grp-2s apparently contain more wind-sea energy than at the other sites. Finally, the unique grp-3s at sites P02 and P03 are both dominated by wind-sea systems, confirming the higher $f_p$ values found in Figure 4.

In association with the wave climate pattern mentioned in Section 2.1, further information regarding the system groups can be revealed. Without consideration of the northwesterly swell generated in the Northern Hemisphere, westerly swell arriving at the selected sites could only be generated at the northern border of the computational grid under the influence of equatorial westerlies. Thus, the swell in the grp-1s is from the artificial boundary (Equations (1)–(3)), and because the westerly energy is input at fixed locations and the swell has traveled a fixed distance on arrival at the sites, the peaks of the swell can be the most concentrated in terms of frequency and direction. Similarly, according to the propagation direction, it can be inferred that the SSW and SW systems in the grp-2s are mainly swell generated by storms in the 40°–60°S region of the South Atlantic. However, when such swell propagates northward, some wind-sea components might become involved because of the southeast trade winds blowing across the South Atlantic. Owing to the obstruction of the Angola shore, only wind-sea components from the S direction can arrive at sites P04–P06. Because the locations of P01 and P06 are relatively further from the land, more wind-sea components from the south can arrive at the two sites; there are even a small number of systems that arrive at P01 from the SSE direction (see Figure 8). Nevertheless, the occurrence of these components is so low that they cannot be classified as an independent group in the second-step partitioning. At P02 and P03, more wind-sea systems can arrive from the S–SSE direction, meaning that independent grp-3s can be identified, which results in a smaller wind-sea fraction in the grp-2s at the two sites.

The box-whisker plots shown in Figures 6 and 7 indicate the spectral width parameter $\epsilon_0$ and the directional coefficient $d_\theta$ at P01–P06, respectively. As smaller values of $\epsilon_0$ indicate narrower band width, it can be found from Figure 6 that the spectral widths of groups with greater swell components, such as the grp-1s at all sites and the grp-2s at P02–P05, are obviously narrower than the spectral widths of both the grp-3s at P02 and P03 and the grp-2s at P01 and P06. The same principle also leads to the different distributions of $d_\theta$ shown in Figure 7; that is, larger values of $d_\theta$ indicate narrower directional spreading, and the groups containing more swell components present more concentrated distributions of energy in terms of direction.

In comparison with the characteristics revealed for each system group, the statistics of the wind-sea fraction, spectral width, and directional spreading parameters obtained from the nonpartitioned spectra are totally different. As shown in the "all" panels in Figure 5, it is evident that at sites where a dominant system group (with absolutely more energy and absolutely more frequent occurrence than other groups) exists, that is, at P01 and P04–P06 where grp-2s are dominant, the nonpartitioned statistics of the wind-sea fraction agree better with those of the grp-2s, whereas at sites P02 and P03, where grp-3s exist and the advantage of the grp-2s is less obvious, the nonpartitioned wind-sea fractions present a "mean" status between that of the three groups. Finally, as shown in the "all" panels in Figures 6 and 7, the nonpartitioned spectral widths and directional spreading are certainly broadest. This is because the width estimation involves multiple systems with energy distributed in different frequency bands and directional regions.

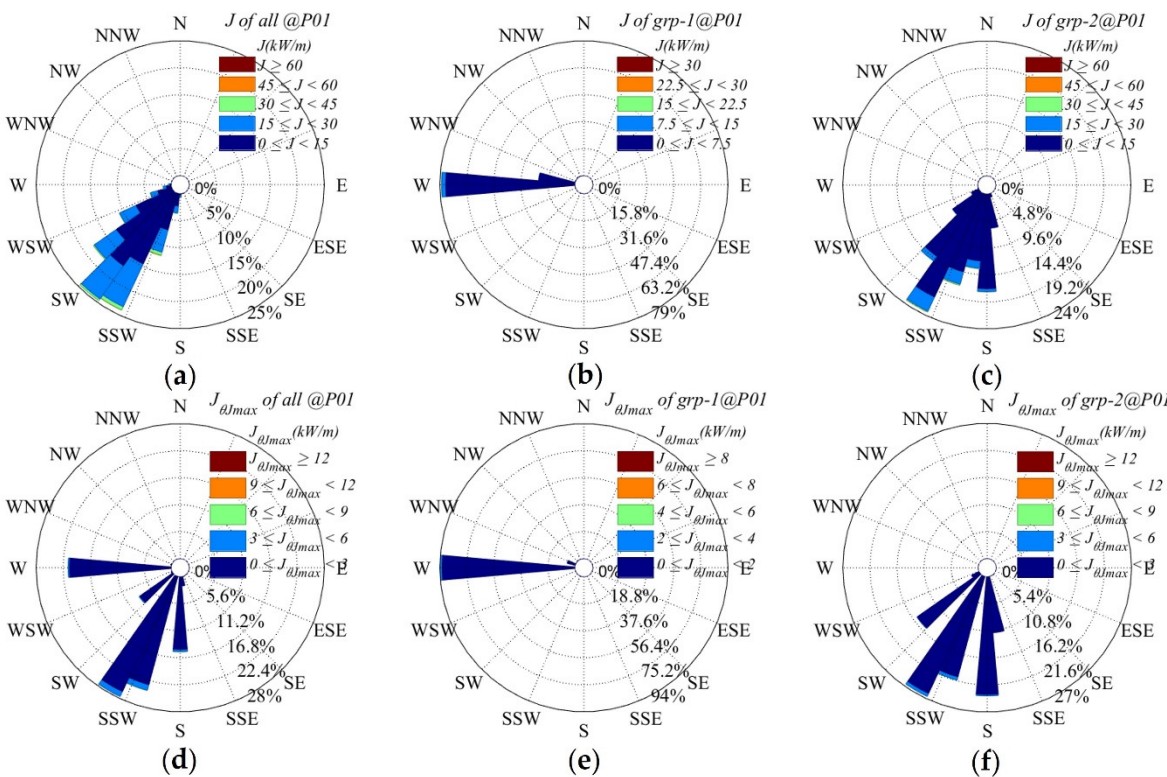

**Figure 8.** Rose plots of (**a**–**c**) $J - \theta_p$ and (**d**–**f**) $J_{\theta Jmax} - \theta_{Jmax}$ joint distributions at site P01. Panels (**a**) and (**d**) exhibit the two distributions of the nonpartitioned spectra, whereas the following columns present the two joint distributions of the grouped systems.

### 3.3. Directionality

Taking the wave energy propagating through P01, P03, and P04 as examples (because similar propagation patterns can be derived correspondingly at P06, P02, and P05), Figures 8–10 illustrate rose plots of the omnidirectional wave power and peak directions ($J - \theta_p$), and thus the maximum directionally resolved wave power and corresponding direction ($J_{\theta Jmax} - \theta_{Jmax}$) at the three sites. In these figures, the panels in the first column on the left exhibit the $J - \theta_p$ and $J_{\theta Jmax} - \theta_{Jmax}$ joint distributions of the nonpartitioned spectra, whereas the following columns present the two joint distributions of the grouped systems.

As shown in panel (a) of each of the three figures, the wave energy is derived mainly from directions between WSW and SW; however, this result might not be correct. As mentioned in IEC2015, peak wave direction $\theta_p$ is incapable of representing the direction of wave energy propagation and is highly unstable; thus, the use of parameter $\theta_{Jmax}$ is recommended. Figure 8d, Figure 9e, and Figure 10d show that for the same nonpartitioned spectra, the prominent directions denoted by $\theta_{Jmax}$ are not only around SW, but also around W and S, and even between S and SSE at P01 and P03. Furthermore, on the basis of the grouped systems (as shown in Figure 8e,f, Figure 9f,h, and Figure 10e,f), it is evident that the grp-1 wave energy is derived solely from the W direction, the grp-2 energy might come from the SW, SSW, and S directions, separately, and the grp-3 energy at P03 comes from the S–SSE direction.

Parameter $\theta_{Jmax}$ might be a better choice to indicate the prevailing propagation direction of wave energy; however, it should be noted that the total wave energy coming from the statistical "most prevailing $\theta_{Jmax}$" is still unknown and cannot be represented by the corresponding $J_{\theta Jmax}$. As introduced in Section 2.3, parameters $\theta_{Jmax}$ and $J_{\theta Jmax}$ are selected from $n$ $J_\theta$ s (in this study, $n = 24$) for a given spectrum and they are most prominent only for the given moment. Thus, the energy potential in the most prevailing $\theta_{Jmax}$ cannot be estimated unless all the $J_\theta$ s (usually 24 or 36) or the series of spectra have

been stored during the wave field simulation, which is generally unfeasible in operational wave energy assessment.

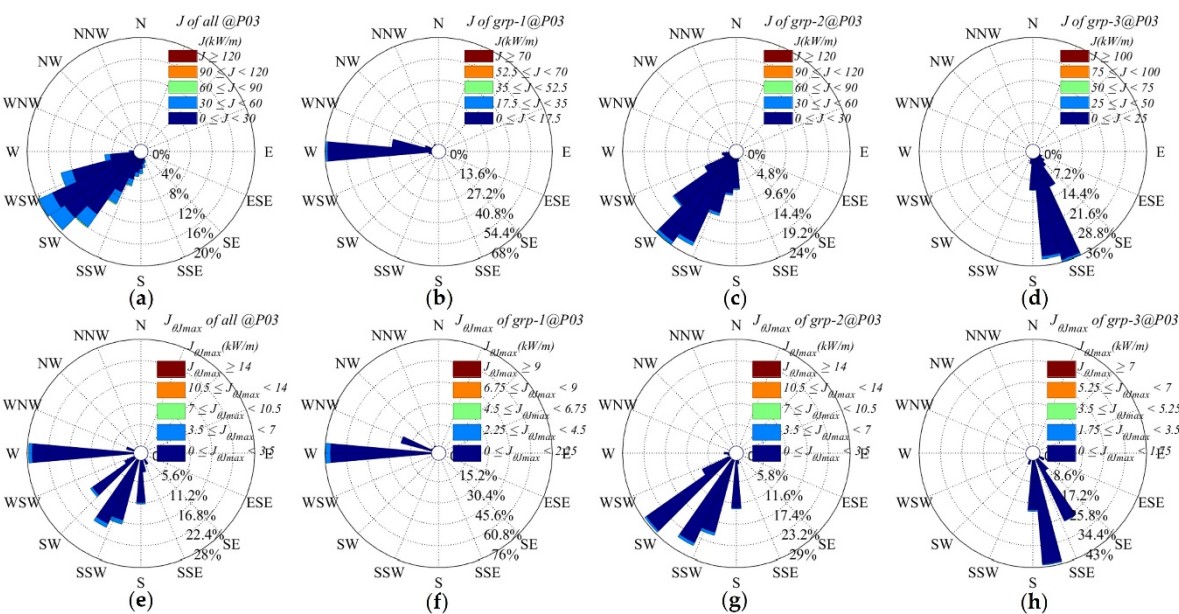

**Figure 9.** Rose plots of (**a**–**d**) $J$—$\theta_p$ and (**e**–**h**) $J_{\theta Jmax}$—$\theta_{Jmax}$ joint distributions at site P03. Panels (**a**,**e**) exhibit the two distributions of the nonpartitioned spectra, whereas the following columns present the two joint distributions of the grouped systems.

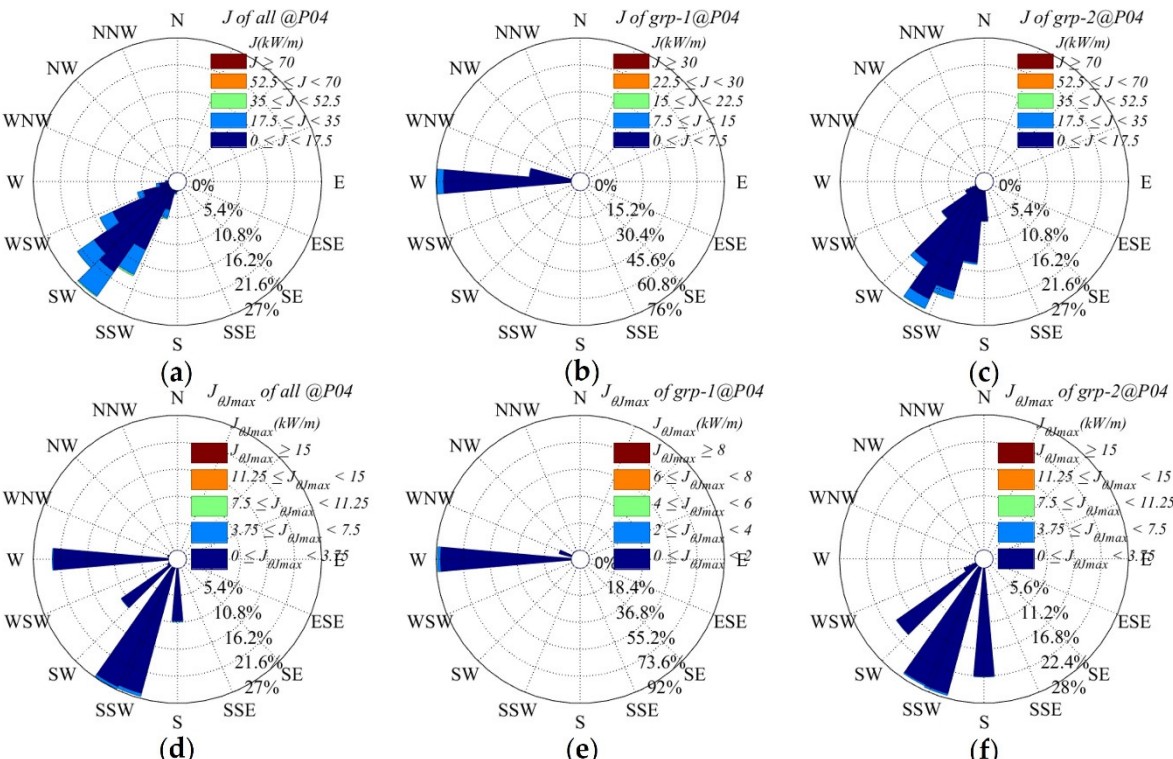

**Figure 10.** Rose plots of (**a**–**c**) $J - \theta_p$ and (**d**–**f**) $J_{\theta Jmax} - \theta_{Jmax}$ joint distributions at site P04. Panels (**a**,**e**) exhibit the two distributions of the nonpartitioned spectra, whereas the following columns present the two joint distributions of the grouped systems.

The deviation of directions indicated by $\theta_p$, which reflects $\theta_{Jmax}$, is caused mainly by the identification of spectral peaks associated with multiple systems. When more than one peak (system) occurs in a spectrum, and $\theta_p$ is still identified as the highest, the energy contained in other systems spreading in other directions can be ignored. With the increase and decrease in the energy contained in each system, the highest peak might suddenly switch from one system to another, which causes $\theta_p$ to be very unstable. For the partitioned spectra, because only one prime peak exists in each partition, the energy distribution in terms of both frequency and direction can be more focused, and the direction of energy propagation can be more consistent with $\theta_p$, as shown in Figures 6 and 7. Therefore, the $\theta_p$ s identified in the grouped systems can confirm the $\theta_{Jmax}$ s well, as shown in Figure 8b,c, Figure 9b,d, and Figure 10b,c. Furthermore, the wave energy potential can then be naturally and physically associated with the statistics of the $\theta_p$ s occurring in each group.

### 3.4. Wave Conditions

The $H_s$–$T_e$ distributions at sites P01, P03, and P04 are shown in Figures 11–13, respectively; again, similar wave environments can be found correspondingly at P06, P02, and P05. In these figures, panels (a) and the other panels exhibit $H_s$ and $T_e$ integrated through the nonpartitioned spectra and grouped systems, respectively. The bivariate distributions are expressed based on 0.5 m × 1 s cells ($\Delta H_s \times \Delta T_e$), and the sea state in the cell located at line $i$ and column $j$ is denoted as $(H_{s,i}, T_{e,j})$. The numbers in each cell indicate the annual average percentage (%) of the occurrence of $(H_{s,i}, T_{e,j})$, and the percentage is calculated as the ratio of the average hours of the occurrence of sea state $(H_{s,i}, T_{e,j})$ per year to the number of hours in 1 year (approximately 8760). The colors in the cells indicate the annual average accumulated wave energy potential (in MWh/m), which is calculated as the product of the hours of occurrence of $(H_{s,i}, T_{e,j})$ occurring per year and wave power $J_{ij}$, calculated based on $H_{s,i}$ and $T_{e,j}$ according to Equation (12). Moreover, the $J_{ij}$ isolines (dashed lines) for the 2, 10, 20, 50, 100, and 200 kW/m energy classes are also presented in each panel.

The significant variability of the sea state among the grouped wave systems can be found in these figures. For these sites shown, groups containing more swell components present sea states with very broad $T_e$ distributions, and the highest value of $T_e$ can extend to 23 s in the grp-1s and grp-2s; however, the sea states presented by grp-3 at P03 have much narrower distribution of $T_e$, and the largest $T_e$ is only 11 s. The sea states that occur most frequently in each group are also different, making the most energic sea state different for each group. Generally, the comparison of groups with more wind-sea components reveals that groups with more swell have the most wave energy potential in $(H_{s,i}, T_{e,j})$ cells with higher $T_{e,j}$, and sea states with higher $H_{s,i}$ occur more frequently in grp-2 than in other system groups.

In contrast to the complex wave environments provided by the system groups, the nonpartitioned sea states appear completely different, as shown in panel (a) of Figures 11–13. In comparison with the wide distribution presented by the swell groups, the $T_e$ distribution of the nonpartitioned spectra is much narrower at each site, and the largest $T_e$ is only 14–15 s. Moreover, the nonpartitioned $H_{s,i}$ certainly occur more frequently in the cells of larger values than in the partitioned ones. Similar to the dominant effect presented by the grp-2s in terms of the statistics of the wind-sea fraction and directionality, the nonpartitioned wave conditions are similar at sites P01 and P04. At P03, where the total contribution of grp-1 and grp-3 is also considerable, the nonpartitioned wave environment presents a "mean" status between that of the three groups.

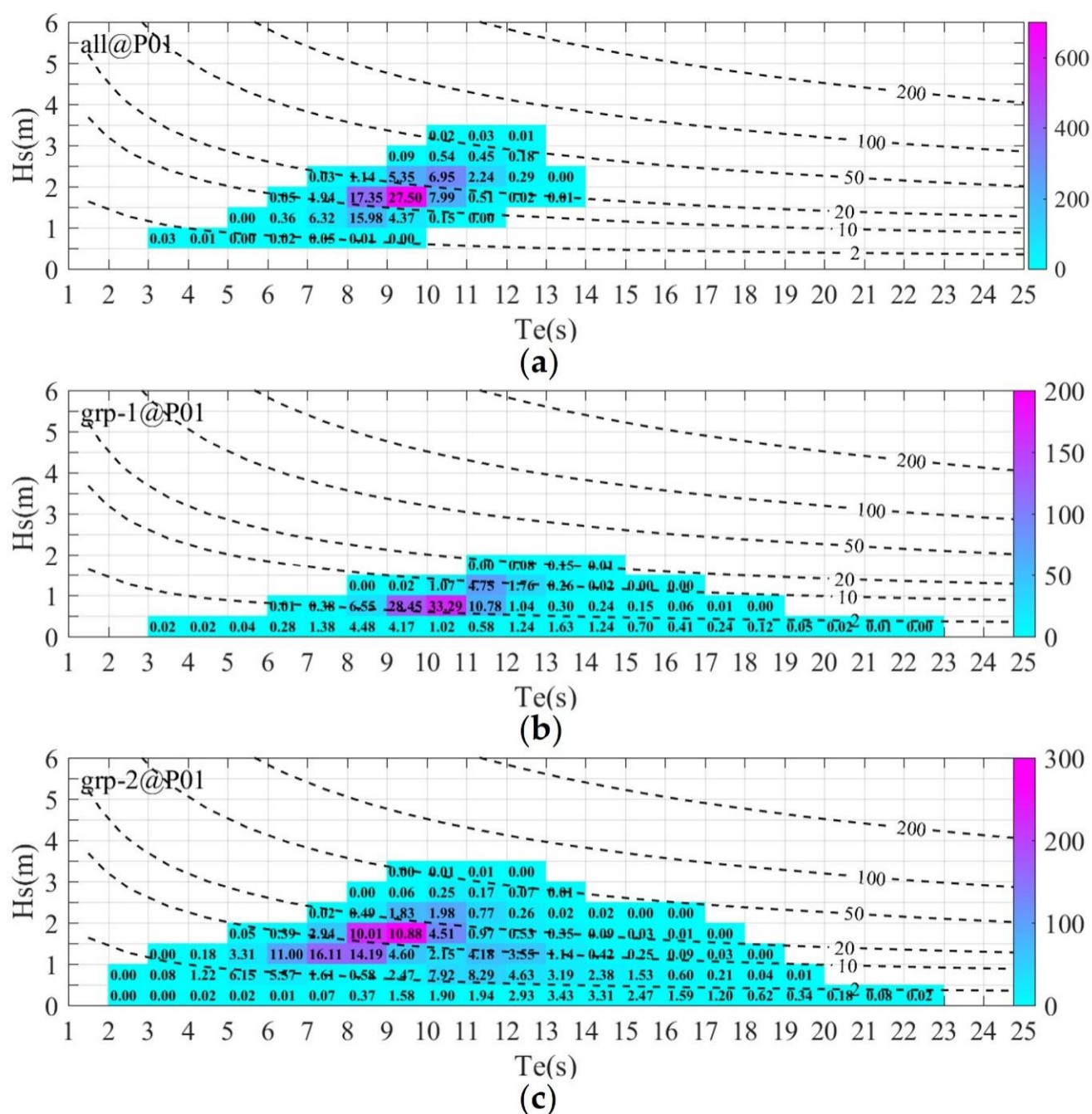

**Figure 11.** Bivariate distributions of $H_s - T_e$ for the nonpartitioned spectra (**a**) and grouped systems (**b**,**c**) at site P01. The bivariate distributions are expressed based on 0.5 m ($\Delta H_s$) × 1 s($\Delta T_e$) cells. The numbers in each cell indicate the annual average percentage (%) of the occurrence of the ($H_s$, $T_e$) combinations, and the colors indicate the annual average accumulated wave energy potential (in MWh/m). Dashed lines indicate the isolines of the 2, 10, 20, 50, 100, and 200 kW/m energy classes.

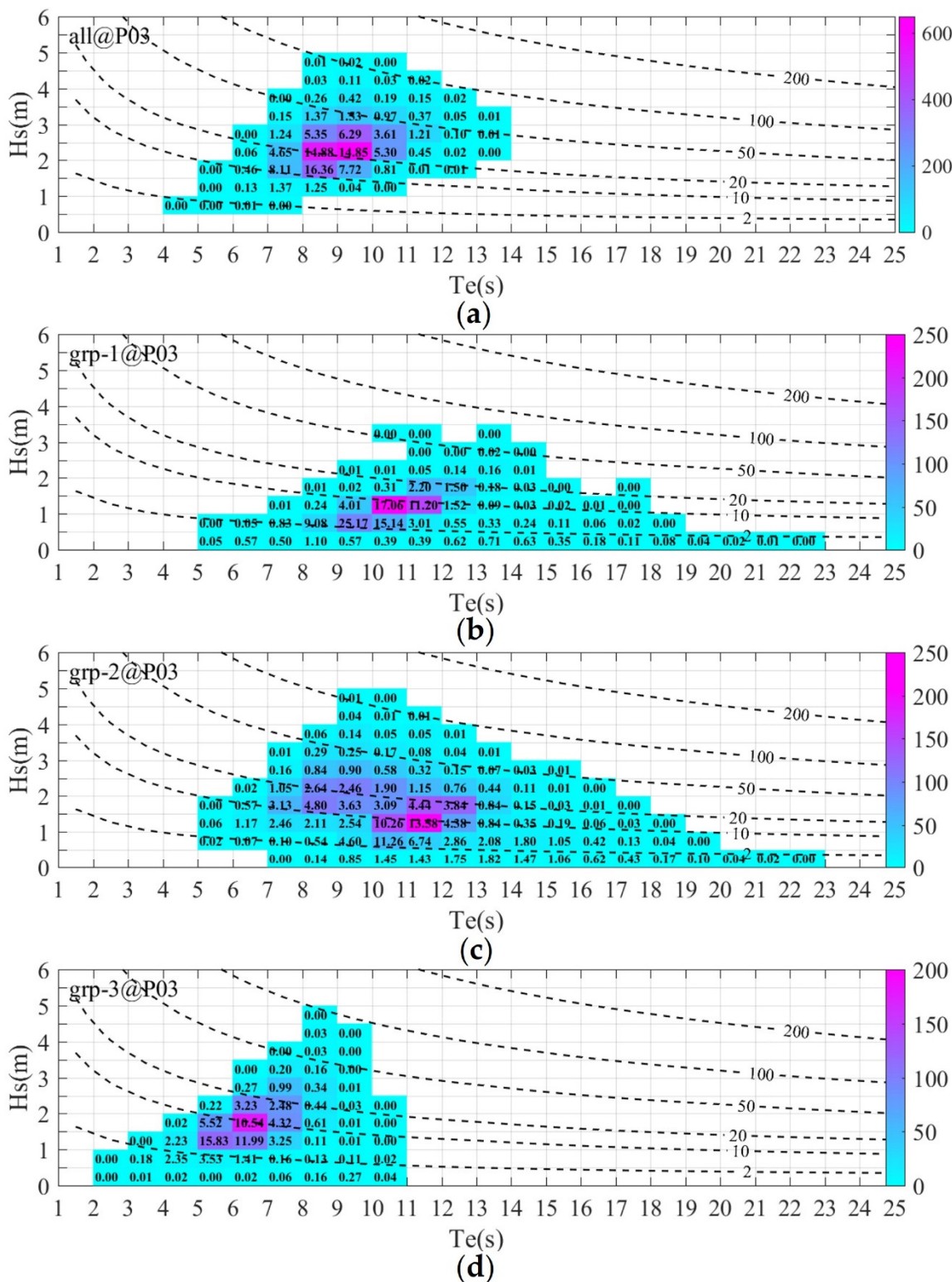

**Figure 12.** Bivariate distributions of $H_s - T_e$ for the nonpartitioned spectra (**a**) and grouped systems (**b–d**) at site P03. The bivariate distributions are expressed based on 0.5 m ($\Delta H_s$) × 1 s($\Delta T_e$) cells. The numbers in each cell indicate the annual average percentage (%) of occurrence of the ($H_s$, $T_e$) combinations, and the colors indicate the annual average accumulated wave energy potential (in MWh/m). Dashed lines indicate the isolines of the 2, 10, 20, 50, 100, and 200 kW/m energy classes.

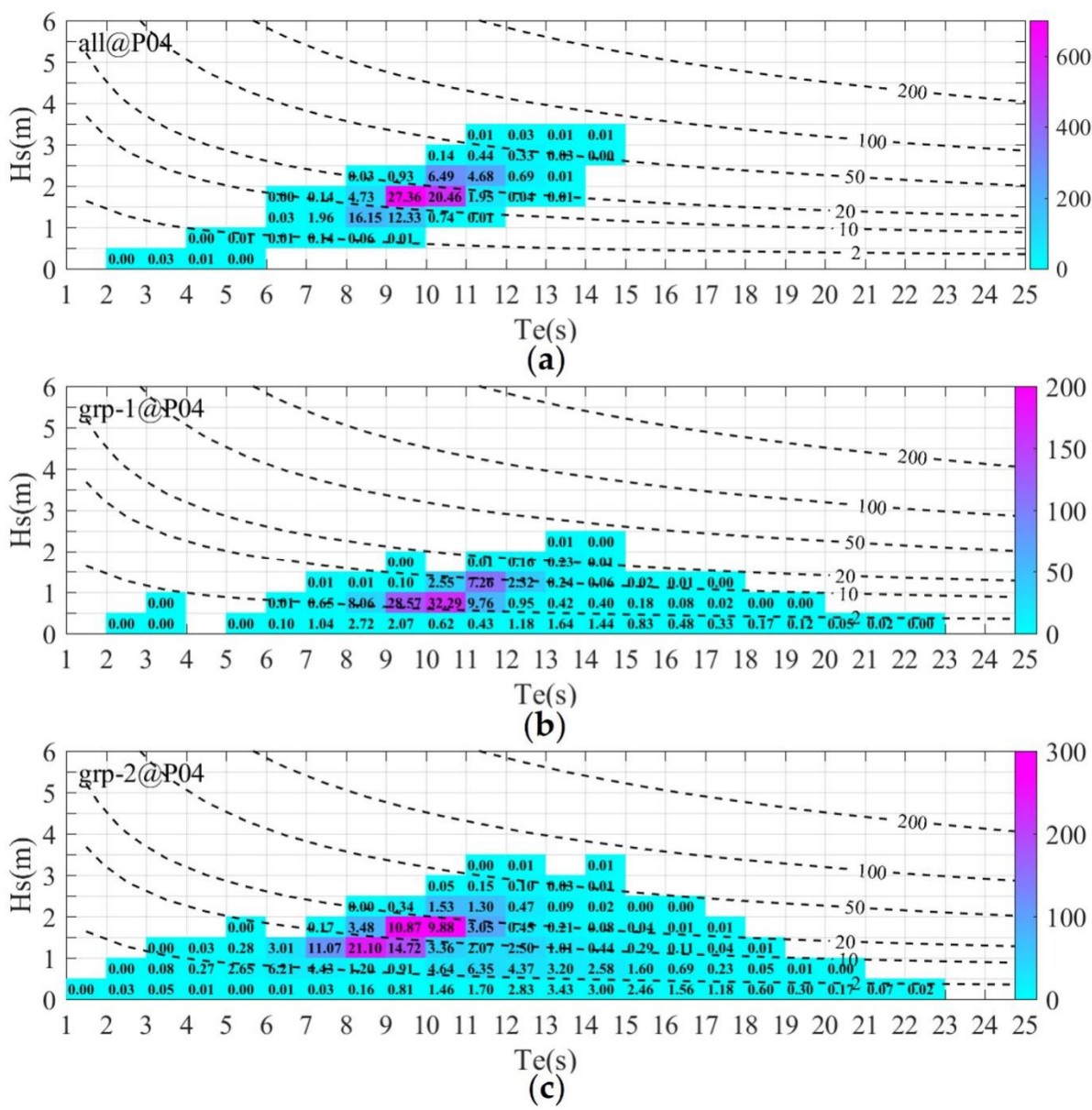

**Figure 13.** Bivariate distributions of $H_s - T_e$ for the nonpartitioned spectra (**a**) and grouped systems (**b**,**c**) at site P04. The bivariate distributions are expressed based on 0.5 m ($\Delta H_s$) × 1 s($\Delta T_e$) cells. The numbers in each cell indicate the annual average percentage (%) of the occurrence of the ($H_s$, $T_e$) combinations, and the colors indicate the annual average accumulated wave energy potential (in MWh/m). Dashed lines indicate the isolines of the 2, 10, 20, 50, 100, and 200 kW/m energy classes.

## 4. Discussion

This paper presents an improved approach to wave energy assessment and characterization, which considers that wave systems are the fundamental components of wave energy, and that grouping long time series wave systems according to their propagation velocity and direction can reflect physical reality and have climatic significance. Therefore, the newly proposed approach can represent wave energy resources in a physical and natural sense, and it is especially suitable for the sea state containing multiple wave systems. Because the characteristics in the aspects of wind-sea/swell components, wave energy directionality, and wave conditions are distinct between the system groups classified in the experimental case, we consider that only one or several of the groups could be harnessed in an envisioned wave farm, rather than all of them. Moreover, the new characterization is implemented based on those widely accepted and reasonable parameters, without in-

troducing any newly defined and unchecked ones. Thus, the comprehensive and detailed information provided by the newly proposed characterization approach could represent a valuable resource for the design and deployment of WECs, the siting of wave farms, and the selection of system groups worth harnessing. Further, it is suggested that this is fundamental to the improvement of WEC performance in real wave energy harnessing.

It is noted that spurious (insignificant) partitions may arise from the "first-step" partitioning procedure, and more such partitions might have been present if the procedure was applied to observed spectra or a finer resolution was used in the spectral space. However, the spurious partitions are generally low-energy noises, and in a real wave energy assessment and characterization, those noises can be easily eliminated by setting a certain threshold in the "second-step" partitioning; for example, partitions whose $H_s < 0.05$ m or energy flux $J < 2$ kW/m could be ignored in the grouping procedure, since the energy contained by those systems (including the spurious ones) is unworthy of harnessing.

It is acknowledged that the wave field modeling in the experimental case demonstrated is not very rigorous, that is, the simulated results are not strictly consistent with the observed data, and the settings at the northern border of the computational grids might overestimate the influence of equatorial westerlies. Nevertheless, these factors do not affect the presentation and elucidation of the new characterization approach proposed in this paper.

## 5. Conclusions

This paper presents an improved wave energy assessment approach which is especially suitable for multiple wave system-coexisting sea states. Traditionally, the characteristic parameters of wave energy are integrated and counted based on the entire spectrum (nonpartitioned). In the newly proposed approach, following the concept of "two-step partitioning" proposed by Portilla-Yandún et al. [62], each wave spectrum in a long time series is firstly partitioned into one or more wave systems (spectral partitions), and all the identified systems are grouped according to the location where their peaks occur in the frequency–direction spectral space. Then, the widely accepted characteristic parameters of wave energy are calculated and counted based on the grouped partitions. To elucidate the new approach, an experimental case was performed with a 20-year series of hourly simulated wave spectra in the Angola offshore area (West Africa). The characteristic parameters, such as wind-sea fraction, spectral width, directionality coefficient (directional spreading), directionally unresolved/resolved wave energy flux, and bivariate $H_s$–$T_e$ distribution, were calculated, counted, and illustrated based on the identified system groups. For comparison, the same parameters were also obtained and counted based on the nonpartitioned spectra, and relevant analyses and elucidation were presented.

The propagation velocity and direction of wave energy contained in the same group are found to be consistent with each other, but the two characteristics among the groups are obviously different. In terms of the wind-sea and swell components in wave energy, propagation directionality, and wave environment patterns, the characterization results of the traditional approach are completely different from those of the new approach. The reason is that when the characteristic parameters are obtained by integrating through the nonpartitioned spectra, the characteristics of the contained systems with less energy and reduced occurrence might be ignored, or the characteristics of some systems that have similar energy and occurrence frequency might be averaged. Moreover, the energy distribution of the partitioned systems can be reasonably focused on both frequency and direction, but a spectrum with multiple systems may present broader spectral width and directional spreading. Finally, although the statistics of parameter $\theta_{Jmax}$, that is, the direction of maximum directionally resolved wave power, can indicate the real prevailing direction of wave energy, it is unfeasible to estimate the corresponding energy potential from that direction. As only one prime peak can exist in each partitioned system, the energy propagation direction of the systems can be more consistent with $\theta_p$ (i.e., the peak direction) and the $\theta_p$ identified in the grouped systems can also confirm the $\theta_{Jmax}$ well. Therefore, the

wave energy potential can be naturally estimated using the statistics of the $\theta_p$ occurring in each group.

The newly proposed approach can provide a new perspective for the studies on wave energy resource assessment and characterization in sea states with multiple wave systems. Further research on improving the WEC design and layouts in wave farms can be performed on this basis.

**Author Contributions:** Formal analysis, D.G. and F.H.; Funding acquisition, D.G. and F.H.; Investigation, X.J.; Methodology, X.J. and Y.Y.; Supervision, Y.Y.; Validation, Z.W.; Writing—original draft, X.J.; Writing—review and editing, X.J. All authors have read and agreed to the published version of the manuscript.

**Funding:** This research was funded by the National Key Research and Development Program of China with grant number 2021YFC3101604, and by the National Program on Global Change and Air–sea Interaction (Phase II)—Parameterization assessment for interactions of the ocean dynamic system.

**Institutional Review Board Statement:** Not applicable.

**Informed Consent Statement:** Not applicable.

**Data Availability Statement:** Not applicable.

**Acknowledgments:** We thank James Buxton [73] for editing the language of a draft of this manuscript.

**Conflicts of Interest:** The authors declare no conflict of interest. The funders had no role in the design of the study; in the collection, analyses, or interpretation of data; in the writing of the manuscript, or in the decision to publish the results.

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
