# Peer review of "An Improved Approach to Wave Energy Resource Characterization for Sea States with Multiple Wave Systems"

_jmse, doi:10.3390/jmse10101362_

Round 1
Reviewer 1 Report
Dear Authors, please find in the attachment my feedback on the Manuscript.

Reviewer 2 Report
The subject of the study is based on the analysis of wind sea and swell separation, wave partitioning, and multiple wave systems, which is a topic of recent focus. The subject is very topical and important. All the details are well organized and presented in a very good language expression. It presents interesting results and presents a new approach. In these respects, I believe that the work is acceptable as is.
Author Response
Response to Reviewer 2
Thank you for your nice comments on our article.
Reviewer 3 Report
There are some grammar mistakes, revise them to improve the English.
The title is too long, modify it.
Author Response
Response to Reviewer 3
We feel great thanks for your professional review work on our manuscript. As you are concerned, there are several problems that need to be addressed. According to your nice suggestions, we have made extensive corrections to our previous draft, the detailed corrections are listed below:
Question 1: There are some grammar mistakes, revise them to improve the English.
Response: We are sorry for the grammar mistakes. Thank you for your reminder. The English has been checked and polished in the revised manuscript.
Question 2: The title is too long, modify it.
Response: The title has been modified as “An improved approach to wave energy resource characterization for sea state with multiple wave systems”
Reviewer 4 Report
In this paper, the authors have studied wave energy flux of grouped waves using a model, that helps them understand 1) components of wave (wind-generated waves and swells, 2) directionality of waves, 3) wave environment pattern. The paper is well written and organized and can be accepted after a revision.
1- I agree with the authors that the main motivation for them is to provide better analysis for analyzing wave energy resources. But I recommend them to explain why they have chosen Angola offshore area. Is there any industrial project targeting this area?
2- Authors have used a third-generation wave model to do their study. Can they briefly name the advantageous and dis-advantages of the model?
3- The water wave equations presented in sub-section 2.3 are well familiar to ocean engineers and oceanographers. But I would like to ask authors to refer readers to a textbook as well. As you know, some readers of JMSE may not be familiar with the field of ocean engineering.
4- In part III of if Introduction, line (60), where author talk about wave condition and their important for performance prediction of WEC, I would like to suggest authors to add Tavakoli and Babanin, 2021, Wave energy attenuation by drifting and no-drifting plates, Ocean Engineering, 236 108717, to their reference list as it shows how the energy of larger waves interacting with a floating body (resembling either sea ice or energy converter) can be dissipated.
5- I would like to ask authors to briefly explain their future plans in Section 5.
Round 2
Reviewer 1 Report
Dear authors,
Thank you for taking the time and draft detailed replies to my comments. Please see the separate document for my replies. Best regards, Reviewer1
